

# imGLAD: accurate detection and quantification of target organisms in metagenomes

Juan C. Castro[1,2], Luis M. Rodriguez-R[1,3], William T. Harvey[2], Michael R. Weigand[3,4], Janet K. Hatt[3], Michelle Q. Carter[5] and Konstantinos T. Konstantinidis[1,2,3]

[1] Center for Bioinformatics and Computational Genomics, Georgia Institute of Technology, Atlanta, GA, United States of America

[2] School of Biological Sciences, Georgia Institute of Technology, Atlanta, GA, United States of America

[3] School of Civil and Environmental Engineering, Georgia Institute of Technology, Atlanta, GA, United States of America

[4] Division of Bacterial Diseases, Center for Disease Control and Prevention, Atlanta, GA, United States of America

[5] Produce Safety and Microbiology, USDA-ARS Western Regional Research Center, US Department of Agriculture, Albany, CA, United States of America

Corresponding author
Konstantinos T. Konstantinidis,
kostas@ce.gatech.edu

## ABSTRACT

Accurate detection of target microbial species in metagenomic datasets from environmental samples remains limited because the limit of detection of current methods is typically inaccessible and the frequency of false-positives, resulting from inadequate identification of regions of the genome that are either too highly conserved to be diagnostic (e.g., rRNA genes) or prone to frequent horizontal genetic exchange (e.g., mobile elements) remains unknown. To overcome these limitations, we introduce imGLAD, which aims to detect (target) genomic sequences in metagenomic datasets. imGLAD achieves high accuracy because it uses the sequence-discrete population concept for discriminating between metagenomic reads originating from the target organism compared to reads from co-occurring close relatives, masks regions of the genome that are not informative using the MyTaxa engine, and models both the sequencing breadth and depth to determine relative abundance and limit of detection. We validated imGLAD by analyzing metagenomic datasets derived from spinach leaves inoculated with the enteric pathogen *Escherichia coli* O157:H7 and showed that its limit of detection can be comparable to that of PCR-based approaches for these samples (∼1 cell/gram).

# INTRODUCTION

Detection of target bacterial species and strains (e.g., pathogens) in environmental samples is a critical step for robust environmental, clinical and biodefense surveillance studies (*Mande, Monzoorul & Tarini, 2012*; *Miller et al., 2013*). A wide range of methods has been employed to target and monitor selected species in air, water, food or clinical samples. Traditional assays include microscopy, culture-based analyses and, in the case of

pathogens, immunoassays that detect antigens expressed by the pathogen. However, these assays are typically cumbersome (e.g., results are available after at least 1–2 days), and cannot typically detect organisms that are resistant to cultivation or novel. Accordingly, culture-independent techniques, including PCR-based amplification tests or sequencing of genomic DNA, have been developed more recently that provide more rapid and, often, more accurate means to diagnose and genotype bacterial species (*Huang et al., 2017*). Advances in sequencing technologies have also drastically improved DNA collection and sequencing from environmental samples. Currently, it is possible to collect DNA samples from the entire microbial community present in a sample, denoted as metagenomic datasets, which provides new opportunities for diagnostics. Nonetheless, several challenges remain to be addressed in order to qualify metagenomics as an everyday tool for the diagnostic laboratory. Due in part to these challenges, many clinical samples entering a public health laboratory remain undiagnosed for the causative agent despite being subjected to a battery of techniques (*Miller et al., 2013*).

Most importantly, assessment of the minimum amount of sequencing required for accurate detection of target bacterial species in a background of a complex microbial community remains challenging. This problem has important practical applications in environmental and clinical surveillance studies as keystone ecosystem organisms or pathogens may not be among the abundant taxa *in situ*. Detection limits vary depending on the sequencing effort and technology (e.g., read length), and the complexity of the microbial community sampled, i.e., the number and relative abundances of the species present in the sample and their relatedness to the target species. In most cases, these parameters or their effects on the limit of detection remain inaccessible. Experiments with increasing amounts of target DNA added to environmental samples have been performed in the past to empirically establish detection limits (e.g., *Be et al., 2013*). However, a theoretical framework to establish limit of detection based on bioinformatics analysis of metagenomics is still lacking. Furthermore, such empirical approaches are typically cumbersome, and specific to the system tested.

Several methods to evaluate presence or absence of bacterial species based on best match or Bayesian analysis of read mapping patterns against a reference collection of genome sequences such as Pathoscope or Sigma (*Ahn, Chai & Pan, 2015*; *Hong et al., 2014*) have been recently developed. Additionally, taxonomic profilers such as MetaPhlAn (*Segata et al., 2012*; *Truong et al., 2015*) or MetaMLST (*Zolfo et al., 2017*) employ species- or strain-specific genetic markers to identify the different members of the community. However, these approaches rely on Single Nucleotide Polymorphism (SNP) pattern differences against reference genes/genomes, which are difficult to robustly determine, especially in cases of low abundance (i.e., not enough reads available to reliably call SNPs). Importantly, no available tool can detect organisms that are not part of a reference genome database, and most tools are not easily adaptable to include new target genomes as references (e.g., the tools require re-computation of the -typically large- training datasets or reference database to include new target organisms). Further, it is not clear how the co-presence of relatives of varying relatedness to the target organisms in the sample, as often is the case of

environmental samples, affects detection ability and whether or not strain-level resolution can be achieved.

While these previous studies highlighted the challenges associated with accurate detection of genomes in metagenomic datasets, they also provided hints for possible solutions to the problem. More specifically, the detection problem can be framed as a classification problem based on two categories: a metagenomic dataset is designated as positive if the target genome is present; conversely, a negative dataset does not include any sequence originating from the target. Thus, training sets with positive and negative datasets could be used to train a classifier for reliable target detection. Likewise, the taxonomic profilers have shown that some regions of the genome can be used to reliably infer presence of a target, at the species or even sub-species (strain) levels, while other regions are not diagnostic enough. Therefore, the classification problem has two important parameters. One parameter is sequencing depth, i.e., how many times each base of the genome is sequenced or covered by sequencing reads, directly related to the relative abundance of the target organism in the sample and thus, the limit of detection. The other parameter is sequencing breadth, i.e., what fraction of the genome has to be sequenced, after removing (masking) regions of the genome that are not diagnostic enough, for reliable detection. By determining the minimum recovered fraction of the genome needed for detection, reliable detection can be established even in cases where species-specific genes are missing; for instance, due to incomplete sequencing or assembly of the target genome from the metagenomic dataset as an effect of low relative *in-situ* abundance.

Here, we present imGLAD (i*n-silico* metagenomes for Genome Low- Abundance Detection), a new pipeline that incorporates a training classification step with positive and negative datasets as outlined above, and several computational optimizations to address the abovementioned limitations. Application of imGLAD to metagenomes derived from samples of known composition (mock) showed that it can reliably detect target organisms of interest in a background of closely related co-occurring relatives and frequently outperforms other methods.

## MATERIAL AND METHODS

### Overview of the imGLAD pipeline

imGLAD assumes that reads of a metagenomic dataset originate at random from all regions of the genome. Thus, the fraction of the genome that is recovered in the dataset (sequencing breadth) as well as the number of times each region is sequenced (sequencing depth), both depend on the abundance of the organism in the community. Highly conserved regions (e.g., rRNA and tRNA genes), as well as regions resulting from recent horizontal gene transfer (e.g., transposase and integrase genes), can recruit reads from other non-target genomes and misleadingly increase the value of sequencing depth (and hence, estimated relative abundance) in some datasets depending on the gene composition of the organisms present. To address this problem, we developed a framework to identify which fraction of a target genome corresponds to reads that belong to the target and what fraction is the result of spurious matches. This framework has two steps: initial training and subsequent
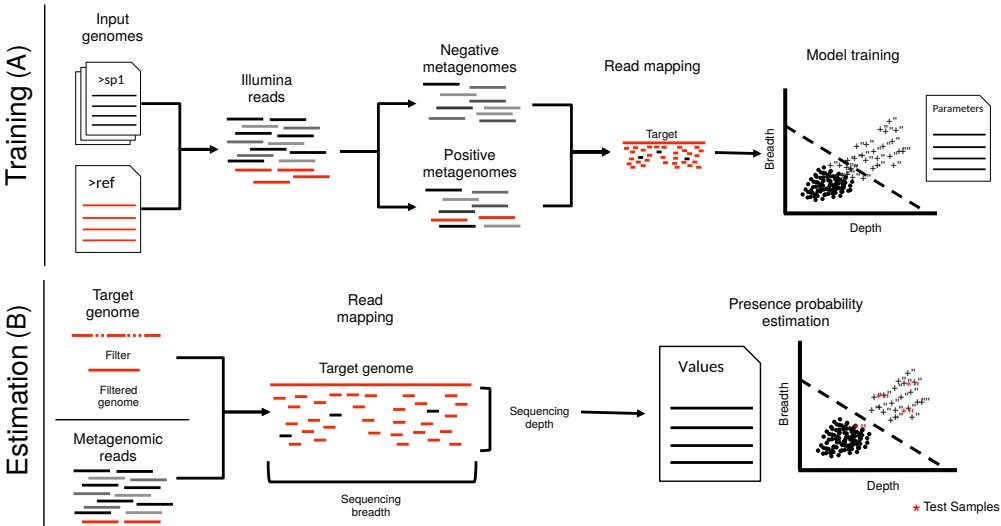

**Figure 1** **Schematic representation of imGLAD's pipeline.** imGLAD has two main components. (A) The first part (training) consists of a learning procedure, in which a set of *in-silico* generated datasets are fitted through a logistic model that aims to separate positive from negative datasets. For this, a database of 200 genomes is used to generate the simulated Illumina reads of these datasets. Reads simulated from the target genome are then incorporated into half of the simulated datasets. The resulting datasets are marked as positive for training while the other half is marked as negative. Sequencing depth and breadth of the target (reference) genome are calculated for each dataset. A logistic function is then fitted to the data to separate positive from negative examples. The regression parameters are stored for further use. (B) The second part (estimation) consists of estimating the sequencing breadth and/or depth values of the target genome provided by the (recruited) reads of the experimental metagenomes, and comparison of the derived sequencing depth and breadth values to those of the logistic function from the training step.

prediction (Fig. 1). Training set selection can be automatic or user defined. The automatic training generates reads from a randomly selected number of genomes (default is 200 genomes) from RefSeq (*Pruitt, Tatusova & Maglott, 2007*), and builds *in-silico*-generated datasets of about 1 million reads each. Simulated reads from the target genome(s) are then generated in a similar way and added to the former datasets in order to create the positive datasets with decreasing target abundances. Reads from the target genome(s) are omitted for the construction of negative datasets. All other genomes used to create the datasets are sampled in equal proportions (i.e., same relative abundances). The user can also choose the genomes to use to generate the training set (e.g., genomes previously known to co-occur in the same environment). In this case, the construction of the training set will be performed based on these genomes rather than the default genome collection from RefSeq. Simulated Illumina-like reads are generated using ART-MountRainier (*Huang et al., 2012*) with default settings. Simulation of reads from additional sequencing platforms is provided as an option, using also ART-MountRainier. Reads from both positive and negative samples are then recruited against the target genome sequence (reference) using BLAT (*Kent, 2002*). Alternatively, BLAST can be used to improve sensitivity at the expense of computational time (*Altschul et al., 1997*). By default, reads with identity higher than
95% and at least 90% of the read length aligned are selected to calculate sequencing breadth and sequencing depth, after normalizing for the size of the dataset. This level of identity has been shown to capture well the genome-aggregate Average Nucleotide Identity (ANI) typically seen between most currently named bacterial species, i.e., >95% ANI within vs. <95% ANI between species (*Konstantinidis & Tiedje, 2005*; *Rodriguez et al., 2018*) and the sequence-discrete populations recovered frequently in metagenomes of natural habitats (*Caro-Quintero & Konstantinidis, 2012*), although different user-defined cut-offs can be used as well. Members of such sequence-discrete populations show high gene-content and nucleotide sequence similarity among themselves, often -but not always- >95% ANI, and/or lower relatedness (e.g., <90% ANI) to close relatives (reviewed in *Caro-Quintero & Konstantinidis, 2012*). Sequencing depth (SD) is calculated as the number of reads mapping to the genome (N) multiplied by the read length (L) divided by the total length of the genome (G), and sequencing breadth (SB) is calculated as the number of bases covered (B) divided by the total length of the genome, using Eqs. (1) and (2) below, respectively. If the genome consists of more than one contig (e.g., draft genomes), the length is assumed to be the sum of the lengths of all contigs.

$$SD = L * N/G \qquad (1)$$
$$SB = B/G. \qquad (2)$$

A logistic function is fitted to the resulting recruitment data (i.e., SB and SD values or SB values alone; see also below) that attempts to separate the positive from the negative training datasets in terms of sequencing depth and sequencing breadth (the latter two are the variables of the function). In particular, this approach calculates the parameters of the logistic function by computing the error in the training set, i.e., what SB and SD values are observed for the 100 positive vs. the 100 negative training datasets, and modifying the parameters accordingly to reduce the error until convergence is reached. Error is assessed by a log-likelihood maximization via gradient approach, which modifies the parameter values until the error is minimized. Regression coefficients of the logistic equation are calculated for the SD and SB variables as well as for an intercept term and thus, the model estimates three parameters, i.e., SD, SB, and intercept. Final parameters of the model are estimated by default only based on SB (sequencing breadth), as this variable was found to be the most discriminating parameter for positive vs. negative samples (see also below). However, an estimation including SD is also provided as an option in order to produce, in addition to the probability of presence/absence, an accurate estimation of the abundance of the target genome.

## Estimation of the probability of detection and limit of detection

Once the parameters of the logistic function have been determined (above), SB and SD can be used to reliably predict the probability of presence of the target genome in any number of query metagenomes after the reads of the query have been recruited against the target genome and (observed) SB is estimated as described above for training datasets. The

probability of presence is estimated according to:

$$p = 1 - \frac{1}{1 \mp e^{-z}} \tag{3}$$

where $z$ is a linear function of the form $\beta^T t$, $\beta$ represents the regression parameters and $t$ is either a vector composed of the SD (Eq. (1)) and SB (Eq. (2)) or, by default, a one-dimensional variable corresponding to SB. Based on the model parameters (Eq. (3)), it is possible to establish a detection limit for the target genome in each metagenomic dataset analyzed. This limit is defined as the minimum fraction (SB) that needs to be sampled in order to estimate a probability of presence at 0.95. The result is displayed as a black solid line in a 2D plot of SB and SD (e.g., Fig. 2). The SD value observed based on the read recruitment, when corresponding to a probability value equal or higher to 0.95, is then used to estimate the relative abundance of the organism in the sample. The SD corresponding to 0.95 probability then provides the limit of detection in terms of relative abundance.

## Filtering conserved regions

To avoid spurious results from reads mapping on regions of the (target) genome with insufficient diversity (high sequence conservation such as rRNA genes) or frequently undergoing horizontal gene transfer such as mobile elements, the user can create a filter for these regions using MyTaxa (*Luo, Rodriguez & Konstantinidis, 2014a*). This filter is created by predicting genes in the target genome and determining their classification weight using MyTaxa. If the MyTaxa classification score is at the bottom 5% or the gene is not scored (e.g., some hypothetical proteins) the gene is removed from the genome and further analysis. The filtered version of the genome is subsequently used for the model training and probability estimation steps.

## Bioinformatic tool comparisons and tool parameters used

MetaPhlAn V2 (*Truong et al., 2015*) was run with the default settings using Bowtie version 2.2.8 (*Langmead & Salzberg, 2012*) for read mapping. MetaMLST (*Zolfo et al., 2017*) was used with default settings. PathoScope 2.0 (*Hong et al., 2014*) was run with default settings, using the same set of reference genomes that were used to build the training datasets for imGLAD.

Four tests were performed to assess specificity and sensitivity. In all cases, sensitivity was calculated as the proportion of properly classified positive datasets among the total number of positive datasets. Specificity was defined instead as the fraction of correctly identified negative datasets among all negative datasets examined. For the first test, metagenomic datasets were created with similar parameters to the training dataset of *E. coli* (i.e., 100 datasets from RefSeq genomes). These datasets were spiked with seven different concentrations of the *E. coli* genome in order to provide 1% to 7% coverage of the genome (i.e., sequencing breadth). In the second test, Human Microbiome Project (HMP) metagenomes were spiked with reads from the *E. coli* genome in order to provide 1% to 7% sequencing breadth as above. 571 HMP datasets were used for each *E. coli* concentration. In the third test, the datasets constructed in test 1 were spiked with reads from close relatives of *E. coli*, i.e., *Klebsiella* (81% ANI), *Salmonella* (82% ANI), and *Escherichia fergusonii* (92%

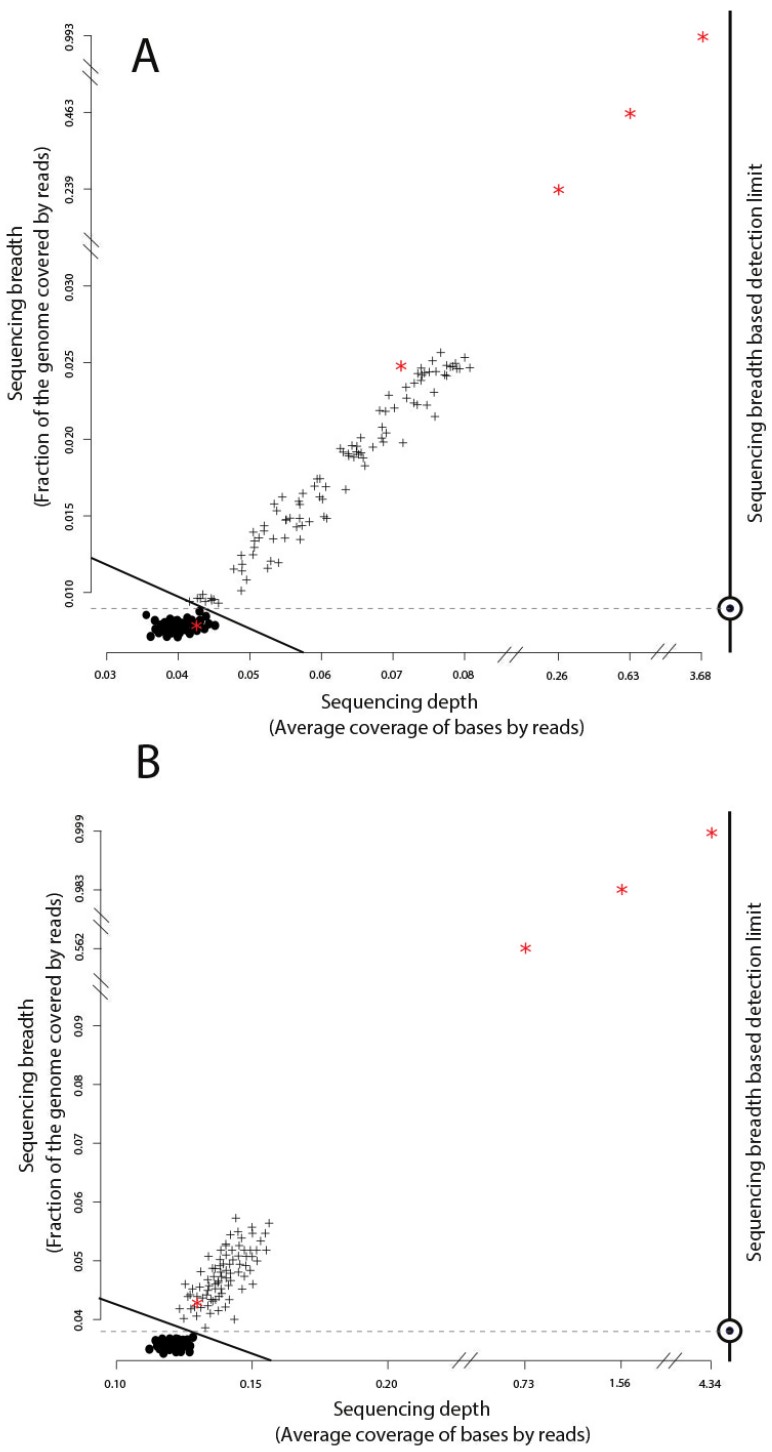

**Figure 2  Identification of target genomes in metagenomic datasets with imGLAD.** Positive datasets (crosses) are separated from negative datasets (dots) through a logistic function (solid line) based on *in-silico* training datasets. (A) Datasets with reads of *E. coli* are separated from negative datasets. (B) Datasets with reads of *B. anthracis* are separated from negative datasets. Red asterisks denote the position of the experimental metagenomes (remaining dots represent *in-silico* generated datasets). Note the differences in scale on the *x*-axes between positive and negative datasets.

ANI), at random concentrations for each genome in addition to the *E. coli* reads. Finally, a test using close relatives, i.e., >95% ANI representing strains of the same species, was performed in the HMP datasets in a similar way as described above for test #3.

## Leaf inoculation experiments to test imGLAD and sample sequencing

Fifty grams of field-grown spinach leaves were inoculated (spiked in) with cells of *Escherichia coli* O157:H7 strain RM6067, a strain linked to the 2006 spinach-associated outbreak in the USA (*Carter et al., 2011*). Three serial dilutions were performed resulting in three inoculation concentrations: 80, $8 \times 10^3$ and $8 \times 10^5$ cells per pellet, plus a control sample with no inoculated cells. Cells for inoculation were obtained from single colonies that were grown overnight, and cell concentrations were determined by enumeration of colony-forming units (CFUs) on LB agar plates. Leaves were subsequently washed, the leaf wash was filtered to remove plant debris, and leaf-associated microorganisms were pelleted by centrifugation at 10,000 g for 10 min at 4 °C. DNA extraction was performed using MoBio UltraClean Microbial DNA isolation kit according to manufacturer's instruction (MoBio).

DNA sequencing libraries were prepared using the Illumina Nextera XT DNA library prep kit according to manufacturer's recommendations, except that the protocol was terminated after isolation of cleaned amplified double stranded libraries. Library concentrations were determined by fluorescent quantification using a Qubit HS DNA kit and Qubit 2.0 fluorometer (Thermo Fisher Scientific, formerly Life Technologies, Waltham, MA, USA) according to manufacturer's recommendations and libraries were run on a High-Sensitivity DNA chip using the Bioanalyzer 2100 instrument (Agilent, Santa Clara, CA, USA) to determine average library insert sizes. An equimolar mixture of the libraries (final loading concentration of 11 pM) was sequenced using a MiSeq reagent v3 kit for 600 cycles (2 × 300 bp paired end run) on an in-house Illumina MiSeq instrument (Georgia Institute of Technology), running the MiSeq control software v2.4.0.4 (MCS). Adapter trimming and demultiplexing of sequenced samples was carried out by the MCS. Additionally, we used metagenomic datasets inoculated with *Bacillus anthracis* DNA, which were made available previously (*Be et al., 2013*).

## McFadden's pseudo-*R²* metric to assess the robustness of the logistic model/function with close relatives

The ability of the logistic model to distinguish between positive and negative training datasets when close relatives of increasing relatedness to the target genome were used in the training step was assessed using the McFadden's pseudo-$R^2$ metric. Specifically, the model determined (fitted) by imGLAD for a certain training dataset was compared to a standard, null logistic model which only contained an intercept variable. Effectively, this null model represented the standard logistic curve centered on the same point as the fitted imGLAD model but without any adjustment to the shape of that curve. Specifically, the metric was defined as:

$$R^2_{\mathrm{McFadden}} = 1 - \frac{\log(L_C)}{\log(L_{\mathrm{null}})} \tag{4}$$

where $L_C$ is the maximized likelihood value for the fitted model and $L_{null}$ is the maximized likelihood value for the null model (intercept only, no covariates). Therefore, if the comparison shows perfect congruence between the two models (pseudo-$R^2$ close to 0 value) this means that the fitted model is not robust but similar to a randomly drawn model. In contrast, when pseudo-$R^2$ approaches 1, this denotes a robust fitted model. Note that pseudo-$R^2$ may not equal 1, even for robust models, because the null model may approximate the fitted model estimated by imGLAD by chance alone in some iterations since it is drawn using the same intercept value. For this evaluation, the genome of one close relative at a time was added to the (negative and positive) training datasets at similar relative abundance (i.e., 10×, to ensured complete genome coverage) as the target genome (*E. coli* strain O157-H7) was added in the positive datasets. The genomes of relatives were sorted into the following groups corresponding to their ANI values to the target genome (%): 90, 95, 96, 97, 98.0, 98.2, 98.4, 98.6, 98.8, 99.0, 99.2, 99.4, 99.6, and 100. No genome was found with ANI value between 99.8% and 99.9% ANI. In addition to these related genomes, a uniform background dataset, which included 200 genomes showing <80% ANI to *E. coli* strain O157-H7, was included to provide positive and negative training datasets of adequate complexity. Successive iterations of imGLAD with the resulting training datasets that each contained one close relative of varied ANI value to the target genome were performed, and models were evaluated using the Eq. (4) above as implemented in the scipy module of Python.

## Availability and dependencies of imGLAD

imGLAD is available through http://enve-omics.ce.gatech.edu/imGLAD/. Source code is available under GNU General Public License v3.0 at GitHub (https://github.com/jccastrog/imGLAD). imGLAD execution requires BLAT or BLAST to be installed, ART and the Python modules "scipy", "numpy", "screed", "statsmodels", and "BioPython" (*Jones, Oliphant & Peterson, 2001*; *Oliphant, 2006*; *Skipper & Perktold, 2010*).

# RESULTS

## Training set for *E. coli* and *B. anthracis*

We evaluated imGLAD's performance on training datasets with *E. coli* strain O157:H7 EC4115, a strain almost genetically identical to RM6067 used in the spinach inoculation experiments, i.e., ~99.97% average nucleotide identity (or ANI), and *B. anthracis* strain Ames as target genomes. The training datasets included closely related (but distinct) species of the same genus with ANI lower than 95% (Fig. 1; see also below for within-species resolution), which corresponds to the frequently used standard for species demarcation (*Goris et al., 2007*) and encompass the sequence-discrete populations recovered frequently in metagenomes of natural habitats (*Caro-Quintero & Konstantinidis, 2012*). Although the predicted detection limit (from the training step) varied slightly for each of the two species, it was always possible to have confident detection (probability of presence >99%) when sequencing breadth was about 0.03 (or 3% of the total genome) or more based on the training datasets used (Fig. 2 & Table 1). The model for *E. coli* was able to accurately

**Table 1** **Samples inoculated with different cell concentrations of *E. coli* (1st column) were classified by imGLAD as present/positive or absent/negative.** The calculated breadth of the *E. coli* reference genome recovered (2nd column) and the sequencing depth (3rd column) as well as the derived probability of presence (4th column) are shown. The estimated relative abundance (fraction of total cells) of *E. coli* in the sample is also shown (6th column). Relative abundance was estimated based on the number of reads mapping on the *E. coli* genome after filtering relative to the total number of reads of the metagenome, assuming all community members had an average genome size of 5 Mbp (similar to *E. coli* genome size) and all mapped reads originated from the *E. coli* cells spiked in (1st column) without losses, i.e., relative abundance = fraction of total reads mapping to *E. coli*. All samples were found positive for presence of *E. coli* ($p$-value = 0.004) except the control sample without inoculated *E. coli* cells.

| Sample | Sequencing breadth | Sequencing depth | *E. coli* presence ($p$-value) | Library size (in Mbp) | *E. coli* relative abundance |
|---|---|---|---|---|---|
| Control | 0.007 | 0.042 | 0.847 | 4,664,749 | ND |
| 80 cells | 0.239 | 0.262 | 0.004 | 4,957,122 | 2.44E–05 |
| $8 \times 10^3$ cells | 0.463 | 0.639 | $7.1 \times 10^{-4}$ | 3,895,441 | 0.0033 |
| $8 \times 10^5$ cells | 0.993 | 3.683 | $1 \times 10^{-5}$ | 3,705,361 | 0.3463 |

separate positive from negative samples (probability of presence > 95%) to a minimal value of sequencing breadth of 0.01 (Fig. 2A).

The logistic models from the training datasets were then applied to metagenomic datasets originating from environmental samples and spiked-in with the target genome (see 'Material and Methods' for details). For the *E. coli* experiment, 100 grams of field grown spinach leaves were inoculated (spiked in) with cells of strain RM6067, a strain linked to the 2006 spinach-associated outbreak in the USA (*Carter et al., 2011*). Three serial dilutions were performed resulting in three inoculation concentrations: 80, $8 \times 10^3$ and $8 \times 10^5$ *E. coli* cells per spinach leaf microbiome (Table 1), plus a control sample with no inoculated cells (no cells were spiked in; although *E. coli* cells might have been present in the background leaf microbial community in low concentration). The resulting samples were sequenced using the Illumina MiSeq short-read technology as described in the 'Material and Methods' section. imGLAD was able to detect the target *E. coli* genome in all samples, even as low as 80 cells (Table 1). For the negative control, imGLAD provided values of sequencing breadth (0.007) and sequencing depth (0.042) that were consistent with the values of negative samples in the training set, i.e., the target genome was not present at the limit of detection of the approach (Fig. 2A; $p$-value for presence: 0.847, Table 1). The matching reads in this case probably originated from natural *E. coli* populations present on the spinach leaves at low abundance or close relatives and/or spurious matches (Fig. S1).

The *B. anthracis* datasets were made previously available by Be and colleagues, and consisted of a soil microbial community DNA sample spiked with known quantities (genome equivalents) of DNA of *B. anthracis* strain Ames (Table 2), and sequenced using the Illumina GA-II technology (*Be et al., 2013*). A training set for *B. anthracis* was built in a similar way to the *E. coli* set; however, genomes that belong to *Bacillus cereus* were excluded from the training dataset in this case as they show ANI values higher than 95% to *B. anthracis*. Based on the training datasets, a slightly higher limit of detection than the one for *E. coli* was obtained (probability of presence >95%), with a minimum value

**Table 2  Soil samples inoculated with different copies of *B. anthracis* strain Ames genomic DNA (1st column) were classified by imGLAD as present/positive or absent/negative.** The calculated breadth of the *B. anthracis* reference genome recovered (2nd column) and the sequencing depth (3rd column) as well as the derived probability of presence (4th column) are shown. Samples with a number of genome higher or equal to 100 genomes were classified as positive samples. Samples with one and 10 genomic copies were indistinguishable from the negative samples of the training set.

| Sample | Sequencing breadth | Sequencing depth | *B. anthracis* presence (*p*-value) |
|---|---|---|---|
| 1 Genome | $1.56 \times 10^{-3}$ | $2.0 \times 10^{-3}$ | 0.999 |
| 10 Genomes | 0.001 | 0.003 | 0.998 |
| 100 Genomes | 0.039 | 0.128 | 0.002 |
| $10^3$ Genomes | 0.562 | 0.732 | $1.4 \times 10^{-3}$ |
| $10^4$ Genomes | 0.983 | 1.563 | 0 |
| $10^5$ Genomes | 0.999 | 4.34 | 0 |

of sequencing breadth of 0.039 (Fig. 2B). Among the six samples tested, a significant probability of presence ($p > 99\%$) was obtained in samples with 100 (3.8% of the genome recovered), 1,000 (56.2%), $10^4$ (98.3%), and $10^5$ (99.9%) *B. anthracis* genome equivalents. The samples with lower genome copy number (one and 10 genomes) were not identified as positive. Manual inspection of the number and position of matching reads to the *B. anthracis* reference genome in the latter two datasets revealed about 2,000 reads for the 10 genome copy dataset and about 4,000 reads for the one genome copy dataset, i.e., more reads were obtained with the lower abundance dataset, indicating spurious matches (each dataset was on average 5.6 Gbp in size). Further, the reads were concentrated in a few regions of the genome (not randomly distributed), which was indistinguishable from negative datasets (Table 2, Fig. 2B, and Fig. S2). Thus, it appears that the *B. anthracis* genomes might not have been sequenced adequately in the low copy number datasets. This interpretation is also consistent with the fact that Be and colleagues employed a phi29-based, whole-DNA amplification method that may have resulted in biased sequencing (e.g., only a few, not enough diagnostic regions of the genome were amplified and sequenced), especially in the low *B. anthracis* genome concentrations. Further, our results are also consistent with estimates that 100 Gbp or more are required to cover the complete genome diversity within typical soil microbial communities as described previously (*Luo et al., 2014b*) and the conclusions of the original study by Be and colleagues.

## Comparison to other tools

We compared the performance of imGLAD with other available platforms that can be used to identify the taxa present in the sample. It should be pointed out, however, that these tools do not target a specific organism/genome of interest but instead assess the total microbial community composition and thus, their objective is slightly different than imGLAD's. Nonetheless, we were able to obtain meaningful results by comparing imGLAD with popular tools for these purposes such as MetaPhlAn, MetaMLST, and Pathoscope, which illustrated the advantages of imGLAD. In the *E. coli* and *B. anthracis* metagenomes described above, imGLAD provided higher sensitivity than other tools, especially at low levels of sequencing breadth. i.e., the proportion of properly classified

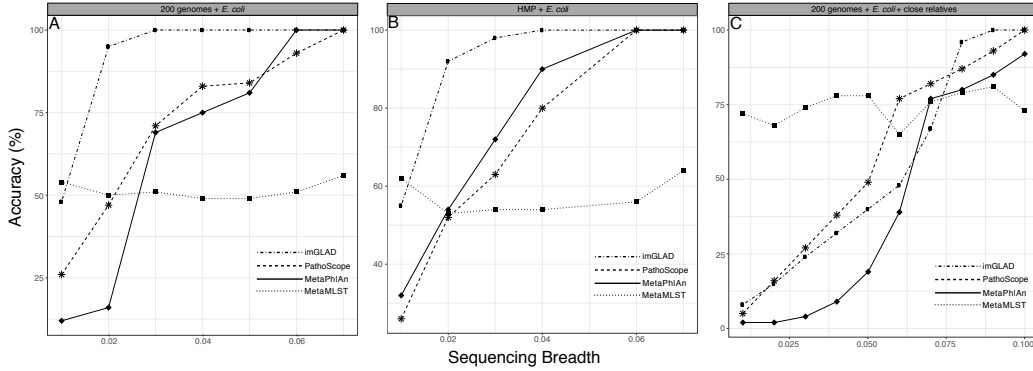

**Figure 3** **Performance of imGLAD in comparison to Pathoscope, MetaPhlAn and MetaMLST.** (A) *in-silico* synthesized datasets from 200 RefSeq genomes were spiked with *E. coli* EC11 reads at different abundances (reflected by sequencing breadth, *x*-axes) to test the sensitivity of imGLAD (*y*-axes), i.e., the proportion of properly classified positive datasets among the total number of positive datasets. (B) Similar comparisons based on 571 datasets from the HMP project, which did not contain any *E. coli* signal and were spiked with different concentrations of *E. coli* EC11 reads. (C) imGLAD was evaluated in the same datasets as in (A) but this time the datasets included, in addition to the RefSeq genomes, 10 *E. coli* genomes with ANI ranging between 95–98% to the target *E. coli* EC11 genome spiked in at the same concentration (i.e., 0.3×). Note that as sequencing breadth increases the sensitivity of the prediction is higher for all tools tested, with the exception of MetaMLST that requires at least 2× sequencing depth for robust detection (see text for details). However, imGLAD can effectively classify samples at 100% sensitivity (as positive samples in this case) with a sequencing breadth as low as 0.03 (i.e., 3% of the target genome recovered) or less, whereas the other tools show lower sensitivity at these levels in all cases evaluated.

positive datasets among the total number of positive datasets. For instance, with 2% of the genome covered by sequencing reads in training datasets, imGLAD accurately classified as positive 95% of the datasets, whereas Pathoscope and MetaPhlAn classified only about 47% and 16% of the datasets, respectively. Only when sampling 7% of the genome or more, did these tools yield similar results to imGLAD (Fig. 3A). It should be noted that 7% is more than twice the genome breadth (i.e., 3% of the genome) that imGLAD required to reach 100% classification sensitivity. Comparisons against available tools that require higher target abundance to make confident calls such as ConStrains (*Luo et al., 2015*) were not attempted as this would have been an unfair comparison. A few other tools such as GOTTCHA (*Freitas et al., 2015*) were proven to be too sensitive, e.g., (positively) detecting *E. coli* even in *E. coli* negative training datasets, and thus, were not evaluated further.

Additionally, we used a set of 571 metagenomic datasets of the HMP (http://www.hmpdacc.org/), in which different concentrations of *E. coli* (target organism) reads were spiked to further test the specificity of imGLAD against a naturally occurring background community (as opposed to *in-silico* generated datasets) (Fig. 3B). These datasets were selected because they did not have any detectable amounts of *E. coli* by any of the three tools to confound results. MetaPhlAn, which is optimized for human-associated microbial communities, had better performance when tested against these HMP datasets relative to the *E. coli* or *B. anthracis* datasets mentioned above. However, MetaPhlAn still required at least 5% of the genome to be recovered in order to provide high confidence

(positive) detection whereas imGLAD achieved similar confidence with only 3% of the genome. Hence, imGLAD's sensitivity was superior, especially in cases of low abundance of the target genome(s).

Improved detection was also observed in *in-silico* synthesized datasets that included close relatives (ANI greater than 95% up to 98% compared to the target; for >98% ANI, see below), although a larger fraction of the genome was typically required in these cases (~7%) in order to achieve high specificity and sensitivity by imGLAD. PathoScope and MetaPhlAn required an even higher fraction (at least 10%) of the target genome for comparable specificity and sensitivity (Fig. 3C; Fig. S3 shows similar results but the background metagenome was from HMP instead of the *in-silico* synthesized datasets). In all cases imGLAD achieved high specificity (>97%), i.e., the fraction of correctly identified negative datasets among all negative datasets examined (Fig. S4). In comparison, the other three tools never reached specificity higher than 90% on the same four tests (Fig. S4).

## Filtering of conserved regions

In addition to creating a model using the whole genome, regions of the genome that provide a less reliable phylogenetic signal (e.g., regions that are highly conserved or contain mobile elements; see 'Material and Methods' for details) can be identified by MyTaxa and removed/masked so that the prediction and/or the training steps can be repeated with the filtered genome for more accurate results. Briefly, MyTaxa was used to examine the classification accuracy of all genes in the genome as described previously (*Luo, Rodriguez & Konstantinidis, 2014a*), and genes with classification score at the bottom 5% or not scored (e.g., some hypothetical proteins) were removed (filtered out) from the genome, providing the filtered genome, which was used for the training step and further analysis. Filtering in general, improved the detection limit because reads mapping on masked regions were not counted (Fig. 4). For instance, filtering lowered the minimum sequencing depth required for robust detection from 0.123× (no filtering applied) to 0.061× in the training datasets for *E. coli* (same *p*-value was used in both cases, equal to 0.05). The reduction in sequencing breadth however was not as dramatic as sequencing depth (e.g., 0.014 to 0.009 fraction of the total genome for the same datasets). The larger effect of filtering on sequencing depth than breadth was presumably attributable to the fact that filtering typically removed only a small part of the target genome (i.e., <5% by default settings) that recruited a disproportionally high number of reads encoding highly conserved or frequently transferred genes. This interpretation is also consistent with the sigmoidal relationship between sequencing depth and breadth, which tends to flatten at high values of sequencing depth and becomes linear at lower values (*Wendl et al., 2012*). Hence, filtering with MyTaxa is recommended, in general, provided the target organism is represented in the database.

## Effect of relatedness of co-occurring genomes and strain-level resolution

When building the training set, the user is able to add any non-target genomes that could be relevant for optimizing detection of the target genome such as genomes that are known to be present and relatively abundant in the sample or closely related species that should not

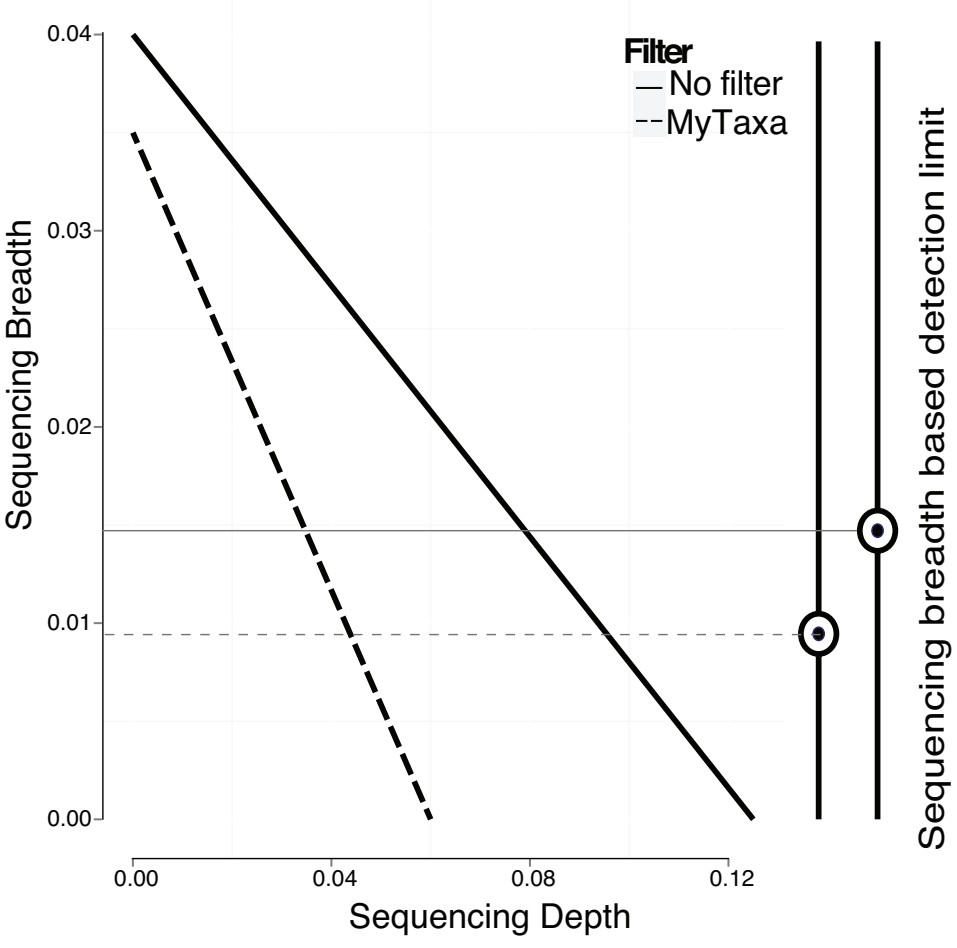

**Figure 4 Effect of filtering of less informative genes by MyTaxa on minimum sequencing breadth and depth.** Genome regions of the *E. coli* target genome were classified by MyTaxa, and regions with low scores (bottom 5%) or no scores because the corresponding genes were not indexed by MyTaxa were excluded from further analysis (filtered genome). Note that detection limit for the filtered genome (dashed line) is lower than the unfiltered genome (solid line).

contribute positive signal (i.e., reads mapping on shared regions of the genome). In general, imGLAD's sequencing breadth and/or depth for positive detection (i.e., the detection limit) was expected to be higher with higher relatedness of the non-target genomes in the training set to the target genome. For instance, we tested different training sets that included relatives at different levels of ANI to the target genome ranging from 80 to almost 100% ANI. Consistent with our expectations, higher sequencing depth and breadth were required for robust detection when relatives showing 95–~98.5% ANI (within species resolution) to the target co-occurred in the training dataset compared to relatives showing 90% or 80% ANI (between species resolution). This was due to the fact that more conserved and/or identical regions were present in the genome of the former relative to the latter. In fact, when co-occurring relatives were members of different species than the target species (i.e., show <95% ANI), imGLAD's limit of detection was very similar to that of the training

datasets without close relatives, i.e., 3% of the genome needed to be recovered for confident detection in most cases. When genomes of the same species were present (i.e., between 95 and ~98.5% ANI), about 10% of the genome was required, depending on the exact genomes considered and their relatedness (Fig. 3C).

When relatives showed higher than ~98.5% ANI (i.e., represented very closely related strains), imGLAD's detection efficiency was increasingly lower up to about 100%, where reliable detection calls were often not possible. For instance, using McFadden's pseudo-$R^2$ metric to assess the ability of the logistic model to distinguish between positive and negative datasets, we found that the model parameters were reasonably well estimated ($R^2 > 0.7$) until relatives showing about 98.5% ANI relatedness to the target were included in the training datasets, at which point the pseudo-$R^2$ fell near 0.5. Increasing the ANI relatedness of the (non-target) relatives further produced pseudo-$R^2$ values either equal to or below 0.5, indicating that the logistic model was not robust in identifying presence of the target in these cases (Fig. 5). Furthermore, as the relatedness increased, the contribution of the sequencing depth metric became less important. In fact, in some training datasets where close relatives with high identity to the target genome (98% ANI or higher) were present and in relative high abundance, the estimated parameters showed high variation during the training step. This resulted, for instance, in a positive slope between sequencing breadth and depth, which was not reliable for estimating relative abundance and detection limit, consistent with the McFadden's pseudo-$R^2$ statistics mentioned above. In these cases, sequencing breadth alone represented a more reliable parameter for robust detection calls. It should be noted, however, that the McFadden's pseudo-$R^2$ evaluation showed that the model based on both sequencing breadth and depth performed better than the one based on sequencing breadth alone (Fig. 5). This was due to the fact that the genome of close relatives was added in the training datasets at the same abundances as the target genome for this evaluation, and that two parameters are typically better than one for model fitting. Hence, the abundance of the target genome relative to that of the close relatives may affect the significance level of the sequencing depth for the logistic function, and the estimation of abundance of the target genome in a query metagenome.

In summary, gene-content differences among the target genome and the co-occurring, non-target close relatives become increasingly more important for robust detection in cases where the non-target genome(s) show increasing genetic relatedness to the target up to about 98.5% ANI, which marks the upper limit of relatedness of the non-target to the target genome for reliable selective detection of the target. Within species (i.e., strain-level) resolution was achievable by imGLAD in such cases, but not when strains shared >98.5% ANI. In the latter cases, the gene-content differences between target and non-target genomes, reflected on sequencing breadth in negative training datasets, was too small to resolve robustly. However, it is important to point out that, in practice, resolution among genomes sharing >98.5% ANI is typically not necessary for most applications since such genomes are typically members of the same sub-clade within species and share highly similar phenotypes, while resolving such closely related genomes would probably require detailed phylogenetic SNP analysis. Hence, training with close relatives is important for more stringent results when needed, especially in cases that the close relatives are known or
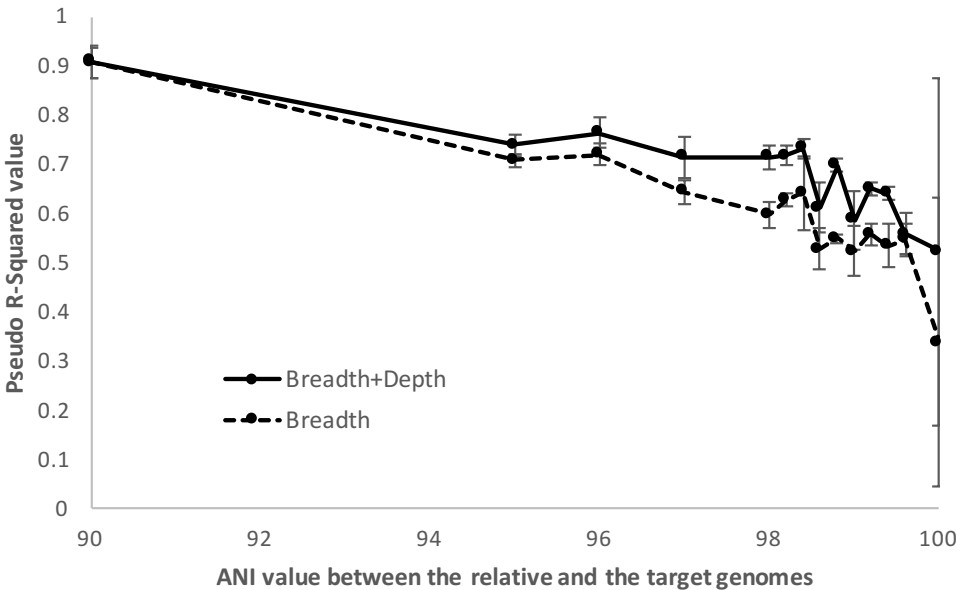

**Figure 5** **Detection limits when co-occurring relatives are present.** Negative and positive examples were constructed using default imGLAD settings except that closely related (non-target) genomes to the target genome at varied levels of ANI (90–100%) were included in the datasets. Results shown are based on *E. coli–E. albertii* genomes related to the target *E. coli* O157-H7 strain. The values shown (*y*-axis) represent the McFadden's pseudo-$R^2$ measure of the confidence level of the regression parameters of the logistic function by comparing the fitted model estimated by imGLAD to a standard, null model that only contains an intercept variable (see 'Material and Methods' for further details). Note that the model performs reasonably well ($R^2 > 0.7$) until the training set shows ~98.5% ANI relatedness at which point the pseudo-$R^2$ falls near 0.5 or below, indicating that the logistic function was not robust.

highly anticipated to co-occur in the same samples with the target organism. When close relatives are not a concern, the default settings of imGLAD should be robust. The default settings are: to create 200 training examples (100 positive, 100 negative) with 1,000,000 simulated 150 bp Illumina reads per dataset; the reads are generated evenly from 200 genomes randomly sampled from NCBI; the positive datasets have additional reads from varying abundances of the target genome; the reads from each dataset are then aligned against the target genome using Blat with thresholds of 95% identity and 90% alignment length in order to build the logistic model.

## DISCUSSION

We presented imGLAD, a novel algorithm that utilizes a logistic model-based learning approach for accurate detection of target bacterial species in complex metagenomes, and for establishing detection limits in a target species- and microbial community-specific manner. By building and analyzing training datasets with decreasing abundances of spiked-in reads originating from the target genome, imGLAD allows for highly reliable calls, while reducing the number of false positives (Fig. 2). Further, and contrary to available tools, imGLAD allows for reliable estimation of the detection limit of the metagenomic sequencing effort applied based on the training datasets of decreasing target genome abundance and a

linear combination of both genome sequencing depth and genome sequencing breadth, or only sequencing breadth. The degree of sequence conservation of the genes of the target genome and their extent of horizontal gene transfer are also taken into account in estimating the limit of detection, which represents a substantial advantage over existing tools in minimizing false-positive calls. The results using both simulated datasets (e.g., Fig. 2) as well as experimental metagenomes (Tables 1 and 2) highlighted these advantages of imGLAD. However, imGLAD is not designed to detect all species present in a sample. Thus, it differs from taxonomic profiling software, and is computationally more expensive, due to the training step, if the goal is to detect more than a couple of targets. Rather, the goal of imGLAD is to provide highly accurate detection of specific, user-provided target species (e.g., pathogens or keystone species), including newly sequenced genomes. Further, imGLAD's logistic model, while computationally demanding to create (e.g., building *in-silico* training datasets), needs to be built only once and can subsequently be used multiple times with different metagenomes. This way, imGLAD could be used to efficiently and rapidly detect several target organisms in an environmental sample (by building a model for each target in advance). For instance, the modeling step typically took 8–12 h on a node with 4 CPUs (2.5 Mzh, 12 Gb memory), whereas the detection (i.e., making a call) step with one query metagenome took 0.5–1 s (once the read recruitment against the target genome was completed; the time required for the latter step varied, depending on genome and metagenome sizes).

A distinguishing strength of imGLAD is the detection of low abundance target genomes. Current tools for metagenomic profiling use specific markers or SNP patterns to identify and classify the species present in the sample (e.g., *Hong et al., 2014*; *Segata et al., 2012*). However, at low levels of abundance, these markers may not be found, and SNPs cannot be called, and in some cases, the SNPs are called incorrectly such as in the case of MetaMLST (*Zolfo et al., 2017*), which requires high abundances (above 2×) to make confident calls and thus, performed poorly in the tests we conducted compared to other tools or imGLAD (e.g., Fig. 3). Our approach is not focused on a particular region of the genome, but instead takes into account the whole genomic context. This provides higher recall while preserving precision (Fig. 3 & Fig. S4). Further, methods based on read assignment depend on the comprehensiveness of their reference database and do not provide high precision when challenged with samples containing closely-related species (*Hong et al., 2014*). Accordingly, the tools evaluated here provided high false positive rates in such cases (Fig. S4), which can be concerning, for instance, in pathogen surveillance studies and environmental samples, where closely related strains of the same species may co-occur. imGLAD can provide reliable prediction even in such cases, although at the expense of a lower detection limit, assuming the close relatives are known and available and, hence, can be used as part of the training step as exemplified in the *E. coli* case above. However, if the query metagenome(s) include relatively abundant, non-target genomes more related to the target genome than any of the genomes used to construct the training datasets, then the predictions of imGLAD (or other tools) might not be highly accurate. In such cases, the user needs to recover the genome sequences of the relatives from the metagenome(s) using genome binning techniques, if the representative sequences are not available otherwise, in order to include them in

the training dataset. The results presented here (e.g., Figs. 3A & 5) provide a quantitative picture of this issue and its consequences on the accuracy of imGLAD as well as other tools.

Electrochemical immunoassays have shown promise in detecting pathogens such as *B. anthracis* or their toxins and can sometimes offer strain-level resolution. The limit of detection of these techniques can, in some cases, be ∼1pg/ml (*Sharma et al., 2016*), which is below the limit of detection of imGLAD (56 pg/ml–560 pg/ml corresponding to 10–100 cells, respectively) based on the *E. coli* spike in experiment on spinach and current best practices for metagenomic sequencing and the samples analyzed here. Thus, immunoassays and culture-based approaches are still more sensitive than metagenomics, at least for highly complex metagenomes such as those of soils (but probably not as much for food and agricultural samples, the human gut or habitats of similar complexity), and could be used in combination with tools like imGLAD for more reliable and comprehensive results. A key advantage of imGLAD is that it has high specificity, which sometimes cannot be achieved by immunoassays or culture-based approaches. It should be noted that imGLAD might be able to offer resolution within species as well, e.g., by including in the training dataset genomes that are members of the same species but show sequence divergence from the target genome higher than that of the sequencing errors (e.g., 99% ANI or less) and/or have substantial gene content differences (which can be captured by the sequencing breadth parameter). Sub-species resolution can also be obtained by analyzing the reads identified by imGLAD as representing the target genome in the query metagenome for their SNP pattern against a collection of genomes related to the target genome, using -for instance- the PathoScope approach (*Hong et al., 2014*) or a (manual) phylogenetic analysis of the reads.

Notably, imGLAD allows one to include new target genomes, including draft assemblies, in the training datasets, with little effort, which may be important for practical applications. Thus, the training step of imGLAD can be optimized with specific targets or habitats in mind such as the human gut and provide comparable, if not better results than tools that are already optimized for these communities. In contrast, most tools available require time and CPU-intensive updates of their reference databases to include new targets. Similarly, imGLAD can be easily optimized for different sequencing technologies as long as the training datasets are produced with reads simulating these technologies. This flexibility of imGLAD is an important advantage because the tenet ''one approach fits all'' does not apply well in the case of microbial detection in environmental samples, which are typically characterized by different degrees of co-occurring (non-target) relatives and are often sequenced based on different strategies nowadays.

It is also important to note that imGLAD's training step can be optimized to evaluate samples of different microbial community complexity, in addition to co-occurring relatives of varied genetic similarity to the target organism and different target genomes. For instance, more complex communities can be simulated in the training step by including a higher number of different genomes in the training datasets (200 genomes by default) and/or with different species abundance distributions, e.g., power law as opposed to equal abundances (default setting). We have also found that training datasets with 200 genomes work well for most natural communities of medium-to-high complexity while increasing the number

of genomes only marginally increased the specificity or sensitivity of imGLAD (Figs. S5 & S6), in general, especially given the extra computational time required. Specifically, our assessment showed that if the richness of the targeted microbial community (i.e., number of 95% ANI defined species or clusters) is within one order of magnitude of the number of genomes used in the training (i.e., one through 2,000 species, for 200 genomes in the training datasets), the estimated imGLAD models are robust. Hence, the default number of genomes ($n = 200$) should work for most microbial communities, and smaller number of genomes could be used for less complex communities (e.g., $n = 100$). The choice of which genomes to use in the training dataset, in addition to just the number of genomes, is apparently also very important for imGLAD's accuracy, e.g., inclusion or not of close relatives as mentioned above (e.g., Fig. 5). Analyzing the target metagenome with profiling tools such as MetaPhlAn and MyTaxa (*Luo, Rodriguez & Konstantinidis, 2014a*; *Truong et al., 2015*) in advance can provide the end user with pertinent information on the taxonomic composition of the target community. This information can guide the selection of the genomes used for the training datasets so that close relatives, when present in the metagenome, can be included for more robust results (e.g., Fig. 5).

We also tested the effect of the size of the training datasets (i.e., library size), in the range of 100 to 1,000,000 reads per dataset, on imGLAD's limit of detection (defined at the $p$ value = 0.05 level as described above). As expected, larger training sets increased the limit of detection because the number of reads recruited by the target genome in such negative datasets (i.e., spurious matches or matches for poor phylogenetic marker genes) increased. However, we found that the increase was only minor overall and, in fact, leveled off around 0.03 sequencing breadth (3%) for 100,000–1,000,000 reads/dataset (Fig. S7). Therefore, the default setting for training dataset (1,000,000 reads) should be robust for most applications. Additionally, we evaluated the accuracy of imGLAD detection calls using the receiver operating characteristic (ROC) curves to measure positive rates and false negative rates over different sequencing breadth. It is important to note that, because the separation of the positive and negative datasets is very sharp (e.g., Fig. 2), the ROC curves were almost perfect for all cases where close relatives of 98% or less ANI to the target, or no close relatives, were present in the sample. In the datasets with close relative showing 98–99% ANI to the target being present, we found some overlap of the positive and negative datasets (Fig. S8), consistent with the results reported above for increasingly more challenging detection with relatives in the 98.5–100% ANI range.

The decreasing costs of sequencing as well as technological improvements in sequencing throughput and read length make it possible to use metagenomics to track specific bacterial populations in time series data or monitor the presence of pathogens in clinical or environmental samples. As the number of studies with a focus on metagenomic datasets continue to increase, the need for fast, reliable and flexible bioinformatics analysis tools to detect and characterize target populations will also continue to grow, particularly in cases where isolation is not possible or is expensive. imGLAD represents an effective way to accomplish this objective and to robustly evaluate the limitations of the underlying sequencing technology or effort. imGLAD's default settings should work for most target microbial communities and genomes, and the results presented here represent a guide for

further optimization depending on the specific goals of the study and the samples analyzed. Therefore, we anticipate that imGLAD will find applications across the fields of clinical and environmental microbiology.

### Funding
This work was supported by the USDA (award 2030-42000-046-10) and the US National Science Foundation (award 1356288). The funders had no role in study design, data collection and analysis, decision to publish, or preparation of the manuscript.

### Grant Disclosures
The following grant information was disclosed by the authors:
USDA: 2030-42000-046-10.
US National Science Foundation: 1356288.

### Competing Interests
The authors declare there are no competing interests.

### Author Contributions
- Juan C. Castro conceived and designed the experiments, performed the experiments, analyzed the data, prepared figures and/or tables, authored or reviewed drafts of the paper, approved the final draft.
- Luis M. Rodriguez-R conceived and designed the experiments, analyzed the data, authored or reviewed drafts of the paper, approved the final draft.
- William T. Harvey performed the experiments, analyzed the data, prepared figures and/or tables, authored or reviewed drafts of the paper, approved the final draft.
- Michael R. Weigand performed the experiments, authored or reviewed drafts of the paper, approved the final draft.
- Janet K. Hatt performed the experiments, authored or reviewed drafts of the paper, approved the final draft.
- Michelle Q. Carter conceived and designed the experiments, contributed reagents/materials/analysis tools, authored or reviewed drafts of the paper, approved the final draft.
- Konstantinos T. Konstantinidis conceived and designed the experiments, analyzed the data, contributed reagents/materials/analysis tools, authored or reviewed drafts of the paper, approved the final draft.

### DNA Deposition
The following information was supplied regarding the deposition of DNA sequences:

All new metagenomic datasets described here are available via NCBI Sequence Read Archive under BioProject PRJNA432668.

## Data Availability

imGLAD is open source software available in GitHub: https://github.com/jccastrog/imGLAD.

## Supplemental Information

Supplemental information for this article can be found online at http://dx.doi.org/10.7717/peerj.5882#supplemental-information.

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
