# Peer review of "imGLAD: accurate detection and quantification of target organisms in metagenomes"

_PeerJ, doi:10.7717/peerj.5882_

## Round 0.1 · original submission · Major Revisions

Having read the reviewers comments and your manuscript I am in agreement that the key areas of the manuscript that need addressing are the introduction and discussion sections. But it is also worth noting that both reviewers have very useful comments and suggestions for the whole manuscript.

·

Basic reporting

Overall the manuscript is fairly well written, although in parts it is made more difficult to read by an overabundance of commas. There are also several instances of incorrect tense being used.

The introduction is too short, and the history of approaches to this problem is not adequately cited. It would also be helpful for the reader to provide a brief description of the problem as it currently approached outside of a purely bioinformatic perspective (e.g. with PCR-based approaches). Then in the discussion it would be helpful to discuss how imGLAD fits alongside these approaches.

The introduction should also briefly describe the presented algorithm in the last paragraph, before the more detailed description is provided in the materials and methods section.

Overall, the problem being approached does not appear to be well defined. The problem is to decide a binary answer to the question “is organism X in sample Y, where Y has a metagenome?”. However, if a very closely related organism is present in the sample (say 2bp different to the target organism across the entire genome), then should the answer be “no”? If the answer should be “yes”, then where is the threshold?

The impact of library size on detection limit is effectively ignored in this manuscript when it is clearly central to this problem. It isn’t clear, for instance, whether the training dataset size and thresholds should be adjusted somehow in accordance with library size. The relationship between genome size and relative cell abundance is also not discussed, when it should be.

In silico is not hyphenated.

The term “sequence-discrete” is neither defined nor widely used in the field, so a description should be given or alternate wording used.

Line 104: “high sequence conservation”: Presumably the authors mean amino acid sequence conservation - this should be stated.

140: The choice of default as being only based on sequencing breadth was surprising given the text and figures, which often described the depth-based approach.

153: extra comma after e.g. (among other occurrences throughout the manuscript)

154: “filtered version”: it is not clear how the filtering was/is done.

194-197: The tools and libraries described should be properly referenced.

201: The inoculation experiments should be briefly described since readers often do not read the methods in detail before the results section.

201: “>99.9%”: A more exact description of the genome’s similarity should be given e.g. the number of SNPs between the two genomes.

216: Spiking in cells rather than DNA makes these experiments more convincing. However, it would be helpful to know the relative abundance of E. coli in this community in percentage terms.

234: These are genome equivalents not genomes, since DNA was spiked in, not cells.

285 and others: depth and breadth should be reported with units.

421: Grammar error. Perhaps “As the number of studies..”

Fig 2: form => from typographical error.
The meaning of the dashed line should be explained.
In (A), the red star which presumably is the negative control invites misunderstanding as it is denoted in the same way as the positives.

Fig 3: Specificity should be presented too.

Experimental design

The sequencing data for the E. coli spike-in experiment should be deposited into a public database such as SRA.

MyTaxa is based on amino acid sequences, but for closely related genomes comparison of nucleic acid sequences would presumably be more discriminating.

It is unclear why BLAST+ or BLAT is used for read mapping, instead of more common read mappers such as BWA or Bowtie that are likely to (1) be much faster and (2) use paired-end information to help correctly place reads. The thresholds for read mapping are also not reported in the manuscript, when they should be.

Runtime information should be included.

While not a prerequisite for publication, the tool would benefit from a setup.py script so that it can be published in PyPI and installed with its Python dependencies without having to manually install each of them.

The tool also seems to require “awk”, and presumably will not work with Python 3 even though the README suggests Python-2.7+. In order to reduce installation issues it may be advisable to use subprocess.check_call rather than os.system so that failing commands are immediately recognized as such.

230: The exclusion of Bacillus genomes seems odd. Inclusion of closely related genomes (but not those that are target genomes) in the training negative set would presumably be of utility to a prediction tool.

233: The relative abundance should be stated.

Validity of the findings

The approach should also be compared with the popular tool described in the following citation because the tool described attempts specifically to reduce the false positive rate incurred by mapping to conserved regions.
Tracey Allen K. Freitas, Po-E Li, Matthew B. Scholz and Patrick S. G. Chain (2015) Accurate read-based metagenome characterization using a hierarchical suite of unique signatures, Nucleic Acids Research (DOI: 10.1093/nar/gkv180)

241-2: The dataset from Be et al was derived from whole genome amplification, and thus concentration of reads in particular areas of the genome is to be expected. Therefore interpreting this as evidence that insufficient sequence was present in the sample is unwarranted. The authors should also mention that whole genome amplification was used for this dataset as this is helpful for interpreting these results.

242-244: The amount of sequencing estimated to be required to sample all diversity within a soil community does not seem necessarily relevant - the abundance of the target organism in the community and depth of sequencing undertaken is all that matters. The authors should give a description of

293: This appears to be an incorrect reference. Further, at such low breadth, it seems intuitively unlikely that sequencing breadth and depth exhibit a significantly non-linear relationship.

325: Strain-level resolution is not assessed in this manuscript meaningfully as it currently exists, and thus a conclusion that it is achievable by imGLAD is not warranted.

398: As discussed above, the richness of a community does not seem specifically relevant.

400: These numbers are entirely speculation, and I would suggest that the choice of reference genomes would be more relevant than their quantity. This is particularly true for soil and environmental genomes, because random choice of genomes is likely to be biased towards medically important lineages.

Additional comments

The use of sequencing breadth is a natural choice as a useful input, even if it is a relatively simple metric. However, depth is more vexed. For instance, in the models described, a low breadth combined with high depth would be regarded as a positive, when this is more likely to indicate incorrect mapping. Intuitively I would expect a model to be less confident when depth is high, not less, after breadth is already taken into consideration.

Reviewer 2 ·

Basic reporting

The submitted manuscript by Castro et al., introduces "imGLAD", a computational tool for quantification of organisms in metagenomic datasets. The manuscript is well written and the literature is sufficient as well as referenced appropriately throughout. The authors provide convincing data to show that their method works as expected with the tested genomes and datasets.

Experimental design

The manuscript meets the aims of PeerJ, and is a solid piece of work with a specific research question that is clear from the title. The background and research methodology are clearly described, however, their introduction needs revised to better describe the scientific problem that imGLAD aims to address. The authors do clearly lay out the existing problems with detection and quantification of microorganisms in metagenomic datasets and point out several limitations that imGLAD is meant to address:

1) emperical approaches are computationally expensive and specific to the system
2) existing profilers rely on SNP pattern differences against a target genome which are hard to determine accurately and are computationally intensive
3) existing tools do not detect organisms that are absent from reference databases
4) existing tools are not easily adaptable to include new organisms (they require recomputation steps)
5) It is not clear how existing tools handle co-occurence of related organisms and if strain-level resolution can be achieved

Unfortunately, imGLAD suffers from some of the same limitations and therefore, it is not really honest to claim that these limitations have been addressed. For instance, "computational expense" is highlighted several times in their introduction as a limitation to existing methods, yet in their discussion (lines 343-349), the authors explicitly state that iMGLAD is also computationally expensive. Furthermore, the argument that emperical approaches are too specific (limitation 1 above) and that existing tools are not adaptable to new organisms because they require new calculations (limitation 4), apply to imGLAD as well. This introduction as-is, unfortunately sets up unrealistic expectations for imGLAD, that ultimately are not met. I would encourage the authors to remove the negativity in the introduction, describe what the existing tools specialize in and then simply describe the specific goal of imGLAD. This would make for a more useful, less combative introduction and would more plainly describe the existing knowledge gap.

Validity of the findings

I find the results to be generally valid, and appreciate the adaptability of the method to discriminate between two closely related and known strains of a similar organism.

My only criticism is that the overall evaluation of their method needs to include receiver operating characteristic (ROC) curves. Reporting Sensitivity alone (Figure 2), "accuracy" (Figure S3) or single point sensitivity / False positive rate (Figure S4 and S5) is not sufficient. Instead, the authors should evaluate the false positive rate and false negative rate over different sequencing breadth to show how the different methods stack up against one another. This is the only fair way to compare the different methodologies, would elegantly replace Supplementary figures S3-S5 and would perhaps be preferable to the existing Figure 2 as well.

---

## Round 0.2 · Major Revisions

The reviewers have raised a number of issues that need to be clarified for the manuscript to be suitable for publication. Could you please address these issues in particular the number of points raised by reviewer 1.

·

Basic reporting

See general comments for the author

Experimental design

See general comments for the author

Validity of the findings

See general comments for the author

Additional comments

The authors have addressed many of my previous concerns, doing further bioinformatic work and most importantly defining the problem more specifically.

On the other hand, I was unable to understand what the results were precisely for this new section, as it is too vague. In the sentence starting on line 356 onwards, for instance, summarises the results when no results have been detailed. Figure 5 provides little help because either the image or legend seems to be incomplete (see below). Also, I was unable to see any methods section corresponding to these results. So while I do not foresee that the summary presented is incorrect, I must reluctantly recommend major revisions in anticipation of a more clear representation of these results.

I apologise for the incomplete sentence left in first referee report.


123: “even richness” is not a correct term, as richness is a whole of community measure of alpha diversity and does not refer to specific lineages.

>> The term “sequence-discrete” is neither defined nor widely used in the field, so a description should be given or alternate wording used.

> We have incorporated a clear definition of sequence discrete to help readers understand the instances where it is mentioned with the appropriate citations.

Abstract, 133, 238: Despite the author’s assertions, I still could not see where “sequence-discrete” was defined.

>>194-197: The tools and libraries described should be properly referenced.

>We have double-checked all instances where a bioinformatics tool or computer language libraries is mentioned and properly included references the first time the tool/library was mentioned.

It appears this effort was at least partially incomplete. The “Bio” i.e. BioPython is not referenced.

>>The tool also seems to require “awk”, and presumably will not work with Python 3 even though the README suggests Python-2.7+. In order to reduce installation issues it may be advisable to use subprocess.check_call rather than os.system so that failing commands are immediately recognized as such.

>Thanks for this suggestion. We have modified the tool to have better interaction with the system and be safer for users.

The README on GitHub still refers to Python 2.7+.

237: “The training datasets included closely related (but distinct) species of the same genus with ANI around 95% or lower (Figure 1; see also below for within-species resolution), which corresponds to the frequently used standard for species demarcation (Goris 2007) and encompass the sequence-discrete populations”. I found this sentence unclear for several reasons. First, if the target ANI threshold is 95%, then why does the training dataset include genomes “around” 95%? Does that set of genomes include any that are >95%? Presumably it should not. Secondly, it appears to imply that a species level is the one and only sequence-discrete taxonomic level when e.g. genera are also that way.

>>285 and others: depth and breadth should be reported with units.

>This is provided now.

The figures remain without units.

Figure 2 legend: form => from (mistake occurs twice).

Figure 2 legend: “Positive datasets (crosses) are separated form negative datasets (dots) through a logistic function (solid line) based on in-silico training datasets.”. Given imGLAD’s default is to use only sequencing breadth and not depth, it isn’t clear why depth+breadth is ascribed as a solid line, where the breadth is only given the less eye catching dashed line. The dashed line is also not described in the legend.

Line 240-243: It is unclear whether the 0.03 figure came from application of the training set, or is just an arbitrary number, as it is not introduced.


256: It is not clear why depth is discussed at all, given it is not a part of the final imGLAD algorithm. This discussion of depth is what led to my original (perhaps incompletely stated) comment:

>>140: The choice of default as being only based on sequencing breadth was surprising given the text and figures, which often described the depth-based approach.

>Indeed, it was a bit surprising for us as well in the beginning, but upon deeper thinking of the problem, this result does make sense and it is well explained in the main text. Accordingly, we have taken no further action to respond to this comment.

If depth is not useful as an input into the algorithm, then why does is it appear prominently in the text, figures and even abstract: “imGLAD achieves high accuracy because it … models both the sequencing breadth and depth to determine relative abundance and limit of detection”.

337: “For instance, filtering lowered
the minimum sequencing depth required for robust detection from 0.123X (no filtering applied) to 0.061X
(p-value < 0.05) in the training datasets for E. coli.” Do these cutoffs refer to the thresholds provided by the training phase at p-value 0.05? It is not clear so that is why I am guessing that.

Figure 5: the legend appears to describe a different figure i.e. it describes 3 lines when only 2 appear to be present in the image.

>>400: These numbers are entirely speculation, and I would suggest that the choice of reference genomes would be more relevant than their quantity. This is particularly true for soil and environmental genomes because random choice of genomes is likely to be biased towards medically important lineages.

>Probably the reviewer has a good point, but we do not know what else we could do about this; neither the reviewer provides any specific suggestions. Accordingly, we have taken no further action to respond to this comment.

The authors might simply wish to tone down their speculation. Perhaps it should be noted I do not feel that a reviewer’s role is to provide constructive suggestions for each and every issue identified in a manuscript, rather, it is the author’s responsibility to find solutions for deficiencies in the manuscript or to dispel the reviewer’s concerns.

417: When using imGLAD to detect species in a new sample, there is a computationally expensive BLAT step. Therefore, I do not see how the runtime could be 0.5-1s as stated. Perhaps the authors mean the application of the logistic model could be applied to a metagenome where the sequencing breadth has already been calculated for a target genome, but in my view that is misleading - the runtime should include the mapping step.

432: “imGLAD can
provide reliable prediction even in such cases, although at the expense of a lower detection limit,
assuming the close relatives are known and available and, hence, can be used as part of the training step
as exemplified in the E. coli case above.” This is overstated - by the authors’ own analyses imGLAD does not provide reliable predictions when the genomes >98% ANI.

484: MethaPlan is misspelled.

The authors use the term “sequencing breadth” to refer to the threshold found in the training step. However, the term is also confusingly used as the unit of depth for simulation-based studies e.g. in Figure 3. This is confusing to a reader - if possible it would be better to only use it in the former circumstance.

The “default settings” are discussed in the results and discussion, but are not defined except in the methods, as far as I could see. Since readers often only briefly read the methods, it would be good to restate what these are in the results section as it is explained what is motivating their choice.

Reviewer 2 ·

Basic reporting

No revisions were necessary.

Experimental design

The revisions were acceptable.

Validity of the findings

I am unsatisfied with the authors' presentation of the ROC curve (S fig 8). They chose to show only imGlad and Pathoscope and it remains unclear why other tools and conditions were omitted. They mention p-values as a prerequisite, but the only numbers really necessary are sensitivity and specificity, both of which are already be calculated. After all, "Accuracy" is defined as the harmonic mean of sensitivity and specificity (legend to Fig S3).Therefore it should be straightforward to display those same values in the form of the ROC curve.This is the standard way to evaluate new methods.

---

## Round 0.3 · accepted · Accept

Thanks you for responding to the reviewers comments and for providing changes which we hope have improved the manuscript.

#